# Perforating Granuloma Annulare with Cysts and Comedones

**DOI:** 10.3390/dermatopathology12020016

**Published:** 2025-05-29

**Authors:** Enric Piqué-Duran, Mikel Azcue-Mayorga, Belinda Roque-Quintana, Odalys García-Vázquez, Antonio Ruedas-Martínez

**Affiliations:** 1Dermatology Department, Dr José Molina Orosa Hospital, 35500 Arrecife, Spain; belinda702@hotmail.com (B.R.-Q.); aruedasmartinez@gmail.com (A.R.-M.); 2Pathology Department, Dr José Molina Orosa Hospital, 35500 Arrecife, Spain; pompoff@icloud.com (M.A.-M.); goladis@yahoo.es (O.G.-V.)

**Keywords:** granuloma annulare, actinic granuloma annulare

## Abstract

A 71-year-old Caucasian woman presented with lesions on both elbows. A physical examination revealed arcuate plaques with raised erythematous edges and central clearing. Comedones and cysts were evident on the border of the lesions. The dermatoscopic view showed the presence of pores, in addition to granuloma annulare changes. The biopsies showed changes according to granuloma annulare, but the granulomas were closely related to comedones and cysts. Furthermore, the presence of elastophagocytosis via multinucleated Langhans-type giant cells was evident. Verhoeff–van Gieson staining highlighted the transepithelial elimination of elastic fibers in the bottom of some cysts. The presence of comedones or cysts is exceptional in granuloma annulare. Only four similar cases have been reported. Although all previous cases showed lesions in sun-exposed areas over photodamaged skin, only our case showed transepithelial elimination of elastic fibers. Diabetes mellitus (DM) could play a role in the pathogenesis of this variant of actinic granuloma annulare, because most cases are associated with uncontrolled DM and the lesions improve after DM is controlled.

## 1. Introduction

Granuloma annulare (GA) is a relatively common skin disorder of uncertain etiology [1]. In the classical localized variant of GA, annular plaques with an erythematous elevated border and central clearing usually occur on the dorsum of the hands and feet. Nevertheless, some other widely accepted variants have been described, including generalized GA, subcutaneous GA, patch-type GA, and perforating GA [2]. Actinic GA or annular elastolytic giant cell granuloma is a controversial condition that we consider a variant of GA. Actinic granuloma is characterized by lesions in sun-exposed areas [2]. Histopathologically, it shows the presence of multinucleated giant cells and is sometimes associated with elastophagocytosis, in addition to the typical changes in GA [2].

## 2. Case Report

A 71-year-old Caucasian woman, Fitzpatrick skin type 2, presented in 2013 with a 2-year history of lesions on both elbows. Her past medical history showed hypercholesterolemia, arterial hypertension, and uncontrolled diabetes mellitus.

Physical examination revealed arcuate plaques with raised erythematous edges and central clearing. Small whitish nodules with some *ostia* on the edge were evident (Figure 1A,B). They were slightly pruriginous. Topical corticosteroids failed to heal the lesions, but the administration of intralesional steroids resolved the lesions *ad integrum*.

Nevertheless, two years later, the lesions relapsed. They had a similar aspect on the right elbow (Figure 1C). However, on the left elbow, instead of small nodules, there were large inflammatory nodules associated with dilated *ostia* and some erosions (Figure 1D). Again, the lesions responded to intralesional steroids.

On dermatoscopic examination, we observed the presence of yellowish-orangish structureless areas associated with white and yellow scaling. A superficial dermal vascular network was also evident. An additional finding was the presence of multiple keratin-filled pores (Figure 1E).

Two biopsies (one from each outbreak) were performed, both showing similar changes. Marked solar elastosis was evident in the papillary dermis, while the reticular dermis showed necrobiotic granulomas. The granulomas were formed by a crown of histiocytes in a palisading pattern and centered by degenerated collagen containing mucin. A perivascular lymphocytic infiltrate was present around some granulomas. The granulomas were closely associated with comedones in the first biopsy (Figure 2A) and with cysts in the second biopsy (Figure 2B). The presence of multiple multinucleated Langhans-type giant cells was evident in the infiltrate in both samples (Figure 2C).

Colloidal iron staining highlighted the presence of mucin in the center of the granulomas (Figure 3A).

Verhoeff–van Gieson staining highlighted the proliferation and agglomeration of the elastic fibers in the papillary dermis, according to solar elastosis, in the surrounding normal skin. They were almost absent from the lesional skin (Figure 3B) but present in the center of the granulomas. Elastophagocytosis by Langhan’s giant cells was easily disclosed (Figure 3C). In addition, transepithelial elimination of elastic fibers was highlighted by this stain at the bottom of some cysts in the second biopsy (Figure 3D,E).

No fungal, bacterial, or mycobacterial infection was demonstrated by Gram, PAS, or Ziehl–Neelsen stains.

Polarized light examination failed to demonstrate foreign bodies.

The blood glucose level was elevated at each relapse—365 mg/dL and 381 mg/dL, respectively—while during non-relapse measurements, the range was 164 to 273 mg/dL. Despite the patient’s treatment with insulin and vildagliptin, her diabetes appeared to be becoming out of control. Furthermore, over time, she developed diabetic retinopathy.

## 3. Discussion

Although the first report of GA is attributed to Thomas Colcott Fox in 1895, the term GA was coined by Henry Radcliffe Crocker in 1902. GA has an incidence of 0.04%. It has a female predilection with a ratio of 3:1 and seems to be more common in Caucasians [3].

Although the etiopathogenesis of GA is unknown, some reports may clarify some of the molecular basis of GA. In 2000, Fayyazi et al. [4] showed a large number of CD3 + T cells expressing interferon gamma (IFNg) and tumor necrosis factor alpha (TNFa). These findings supported a Th1-mediated process via the upregulation of TNFa. Recently, Min et al. [5] evaluated samples of GA lesions, normal skin from patients with GA, and normal skin from healthy controls for a large array of inflammatory markers. They found a different expression of some inflammatory markers in GA skin compared with skin from controls. TNFa, IFNg, and interleukin (IL) 12/23p40 were increased, suggesting the activation of the Th1 pathway. However, other inflammatory markers such as IL4 and IL31 were also increased, suggesting the activation of the Th2 pathway, and JAK3 elevation suggested the activation of the JAK-STAT pathway. Thus, it seems that the Th1, Th2, and JAK-STAT pathways may play a role in the etiopathogenesis of GA. These findings indicate possible treatment targets and may explain the good response of GA to apremilast [6,7] (a phosphodiesterase 4 inhibitor acting on TNFa and IL23p40, among others) and dupilumab [8] (antibody anti-IL4 receptor) in some reports, but the failure of anti-IL23p19 [9]. Wang et al. [10] confirmed the activation of the JAK-STAT pathway in addition to the Th1 pathway, but not the Th2 pathway, through single-cell RNA sequencing and effectively treated five patients with recalcitrant GA with tofacitinib.

Localized GA presents clinically as annular plaques with an erythematous elevated border and central clearing, usually involving the dorsum of the hands and feet and the elbows. Other widely accepted variants of GA have a specific clinical presentation. Thus, in patch-type GA, patients show one or a few patches involving the same locations as localized GA. A subcutaneous nodule, usually on the elbows of pediatric patients, is the way in which subcutaneous GA presents. Perforating GA usually presents itching, umbilicated and keratotic papules, and annular plaques located in the usual places. Finally, disseminated GA usually presents as subtle papules and annular plaques widely extended over the body. Uncommon locations such as the nose [11] or penis [12] have been reported, as well as unusual clinical pictures such as lineal GA [13], pustular palmoplantar GA [14], ulcerative palmoplantar GA [15], giant plaque GA [16], or pseudoxanthoma elasticum-like GA [17].

Histopathologically, GA has two main patterns: (a) a palisading granulomatous pattern, which shows a central core of degenerated collagen that contains mucin and sometimes neutrophilic nuclear dust, surrounded by a crown of epithelioid histiocytes. The presence of perivascular lymphocytes is another common finding; and (b) an interstitial pattern that shows lymphocytes, histiocytes, and mucin interspersed among collagen bundles. However, some challenging findings can be present. In sarcoidal GA, the GA granulomas resemble sarcoidosis. In pseudolymphomatous GA, there is a prominent lymphocytic infiltration that can be confused with cutaneous lymphoma [18,19]. Sometimes, there are so many eosinophils that the diagnosis of GA is difficult. Occasionally, epidermal hyperplasia accompanies GA [20,21].

Individual consideration should be given to actinic granuloma, also known as O’Brien granuloma, Miescher’s facial granuloma, or annular elastolytic giant-cell granuloma. It was first described by O’Brien in 1975 [22]. O’Brien focused his report on the changes in elastic fibers along lesions and their relationship with giant multinucleated cells. He found the absence or slight palisading of histiocytes disposed around clumped elastotic fibers, but the presence of mucin was not investigated. He wrote, “…the two diseases (referring to actinic granuloma and GA) appear related in a manner as yet unknown”. The hallmarks of actinic GA are elastophagocytosis, elastolysis, and the presence of multinucleated cells. However, multinucleated cells are a relatively common finding in GA, as is elastophagocytosis [23]. Even elastolysis has been described in GA [24,25]. In addition, clinically actinic granuloma and GA are indistinguishable. For this reason, we consider actinic granuloma a histologic variant of GA. However, until the etiopathogenic mechanisms of GA formation are confirmed, the debate remains open.

The presence of open comedones or cysts is exceptional in GA. To the best of our knowledge, only four reports have described this association [26,27,28,29]. All cases, including our case, fulfilled the criteria for actinic GA. The age range was 42 to 71 years old, and the female-to-male ratio was 3:2. The five patients had lesions around the elbows, although one case showed an annular plaque on the forehead and some other plaques on the chest. In this case, the comedones were only present in sun-exposed areas [27]. Table 1 shows some details of the cases (Table 1).

The histological findings were quite similar in all cases, showing palisading granulomas in the reticular dermis, closely related to comedones [26,27,28]. Multinucleated giant cells were evident in all cases, in association with elastophagocytosis [27,28], although, in Pensler’s case [29], neither the presence of multinucleated giant cells nor elastophagocytosis was specified. Langhans giant cells were the predominant multinucleated giant cells in our case and probably in Gavioli’s case [28]. The type of multinucleated giant cell was not specified for the rest of the cases. Again, our case shares similarity with Gavioli’s case in terms of the presence of cysts in addition to comedones [28].

Interestingly, our case showed transepithelial elimination of elastic fibers through the cyst wall, a finding that has not previously been described. The transepithelial elimination of material is due to (a) the elimination of a foreign body; (b) secondary to scratching; and (c) the elimination of altered tissue. In our case, the transepithelial elimination of material occurred at the bottom of the cyst, making scratching an unlikely explanation, and the polarized light study was negative. Thus, in our patient, it was probably secondary to the elimination of altered material.

The dermatoscopy was not assessed in the previous reports. Our findings are similar to the changes reported by Errichetti et al. for actinic GA [30], albeit our case showed multiple keratin-filled pores due to the presence of comedones and cysts.

The presence of comedones and cysts could be related to elastophagocytosis. Thus, the loss of the support of elastic fibers could facilitate the appearance of comedones [26,31]. A similar explanation has been proposed for Favre–Racouchot syndrome. However, other factors likely play an important role in cyst formation in these cases, because the presence of comedones and/or cysts is an extremely uncommon event, even in cases of elastolytic GA or actinic GA.

Clinically, our case must be distinguished from halogenodermas, but our patient had no contact with iodine, bromide, or fluoride. Histopathologically, halogenodermas show pseudocarcinomatous epidermal hyperplasia, intraepidermal microabscesses, and dense interstitial inflammatory infiltrate of neutrophils, eosinophils, and lymphocytes, but lack palisading granulomas [32].

From a histopathological perspective, our case could be confused with a ruptured cyst with a secondary granulomatous reaction; however, the granulomas are suppurative or of foreign-body type rather than the palisading granulomas seen in GA. In addition, the material phagocytosed by giant cells is keratin and not elastic fibers.

A foreign body reaction leads to suppurative granulomas, foreign body granulomas, or both, depending on the nature of the foreign body, and not palisading granulomas with mucin deposits in the center. Although it is possible for a foreign body reaction to show transepithelial elimination, cyst formation is not typical. In our case, the results from the polarized light examination, the presence of palisading granulomas, and the clinical picture rule out this diagnosis with confidence.

Nevertheless, a main differential diagnosis must be established in cases of deep fungal infection and mycobacterial infection, either verrucous tuberculosis or atypical mycobacteriosis [33,34]. Verrucous tuberculosis presents with a single verrucous plaque [35]. Infections with Mycobacterium fortuitum and Mycobacterium chelonae usually involve the lower extremities. A traumatism or medical procedure often precedes the appearance of a slowly growing erythematous nodule that can spread in a sporotrichoid pattern [34]. Mycobacterium marinum infection can occur after contact with an aquarium or after bathing in a swimming pool, and appears in the form of an erythematous, sometimes keratotic plaque involving the wrists or hands, and sporotrichoid spreading is possible [34]. In sporotrichosis, there is usually a history of traumatism involving a plant. Although the clinical lesions in our case may resemble a deep fungal or mycobacterial infection, there are important differences. The lesions in our case were symmetrical, had an arcuate form, and contained cysts and comedones, none of which are representative of fungal or mycobacterial infections. Histopathologically, sporotrichosis and atypical mycobacterisois have similar characteristics such as a marked epidermal hyperplasia that often contains intraepithelial abcesses [33]; however, the presence of comedones or cysts without epidermal hyperplasia is not characteristic of this type of infection. In the dermis, there is a dense and diffuse infiltrate composed of histocytes surrounding zones of suppuration [33]. Often, multinucleated Langhans-type giant cells are present. However, well-formed palisading granulomas with mucin in the center are not present in mycobacterial or fungal infections [34]. Moreover, in our case, the Langhans cells showed elastophagocytosis, a finding not present in infection [35]. On the other hand, cutaneous tuberculosis involves tuberculoid granulomas with caseation necrosis in the center, in addition to epidermal hyperplasia. Again, the presence of comedones and cysts instead of epidermal hyperplasia, and the mucin in the center of the granulomas, favor the diagnosis of granuloma annulare with cysts and comedones.

Finally, the clinical presentation and the presence of mucin favors the diagnosis of GA over other palisading granulomas such as rheumatoid nodule or necrobiosis lipoidica.

Interestingly, DM could play a role in the pathogenesis of the appearance of cysts or comedones in GA. The five cases of GA associated with cysts had diabetes mellitus [26,27,28]. Sudy et al. [26] reported the case of a 57-year-old female with GA. Four years after the diagnosis of GA, some comedones appeared on the GA lesions, coinciding with the appearance of DM. The GA and comedones healed when DM was controlled by insulin treatment. Similarly, DM treatment caused the disappearance of the comedones in the case reported by Bhushan et al. [27] Although extremely high blood glucose levels were found in each relapse in our patient, we cannot establish with confidence that the skin lesions improved when the blood glucose levels were lower.

In our case, the response to treatment and evolution were likely to be the same as those in localized GA.

## 4. Conclusions

In conclusion, we presented the fifth case of actinic GA in association with comedones and cysts. This variant of GA may be related to uncontrolled diabetes mellitus, in addition to actinic damage. Interestingly, our case involved transepithelial elimination of elastic fibers, a finding that has not previously been reported. Dermatopathologists should be aware of this atypical variant of GA, which is challenging both clinically and histopathologically.

## Figures and Tables

**Figure 1 dermatopathology-12-00016-f001:**
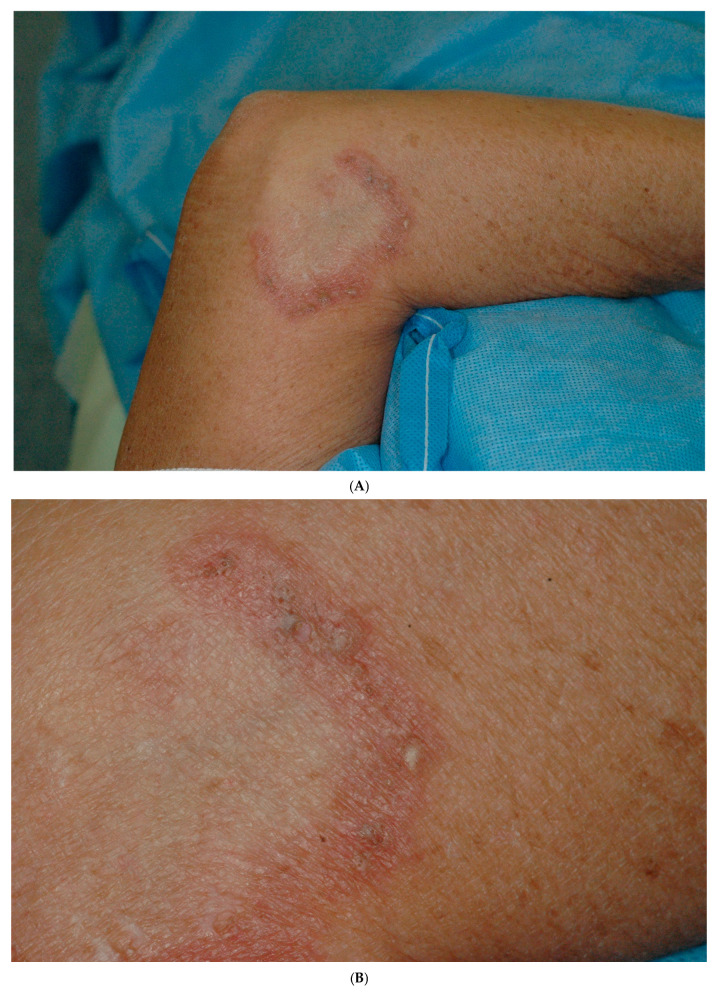
(**A**) Left elbow: Arcuate plaque with raised erythematous edge and central clearing, containing some whitish nodules with some *ostia*. (**B**) Close-up showing where *ostia* of comedones are evident. (**C**) Right elbow: Relapsing arcuate plaque similar to the first episode. (**D**) Left elbow: Relapsing arcuate plaque with some inflamed nodules (**E**) Dermatoscopy: Multiple keratin-filled pores and yellowish-orangish structures associated with white and yellow scaling. The superficial dermal vascular network is evident.

**Figure 2 dermatopathology-12-00016-f002:**
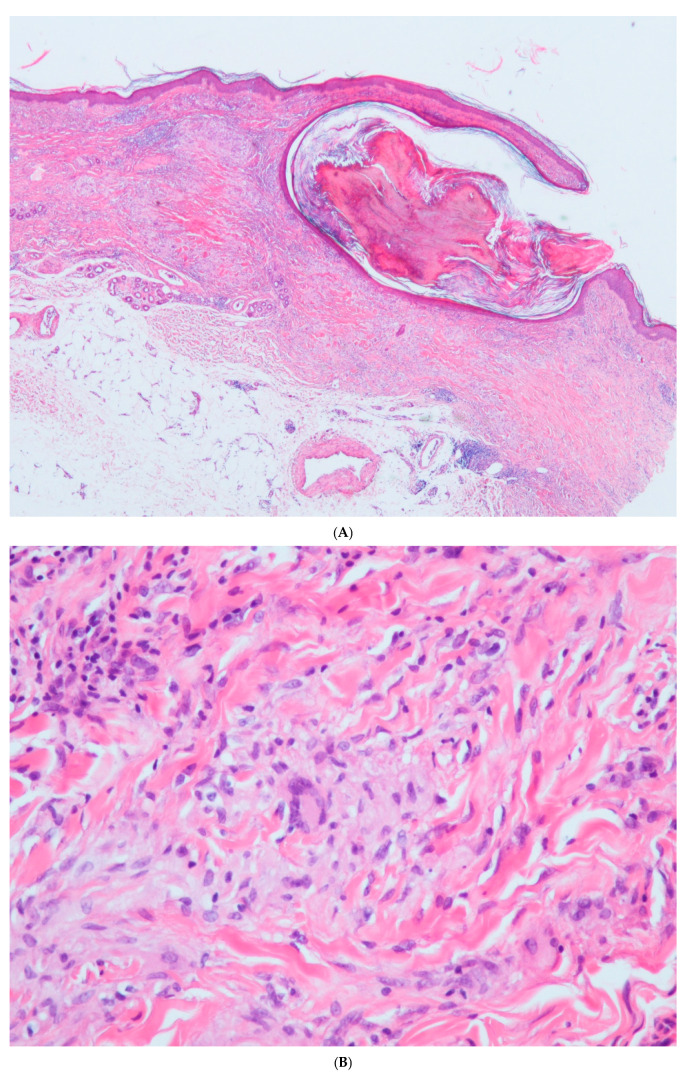
(**A**) Some palisading granulomas in close association with a comedo (H&E ×20). (**B**) Detail of a Langhan’s giant cell in the wall of a granuloma (H&E ×200). (**C**) A palisading granuloma perforating the cyst wall. It is surrounded by a nodular infiltrate of lymphocytes (H&E ×100).

**Figure 3 dermatopathology-12-00016-f003:**
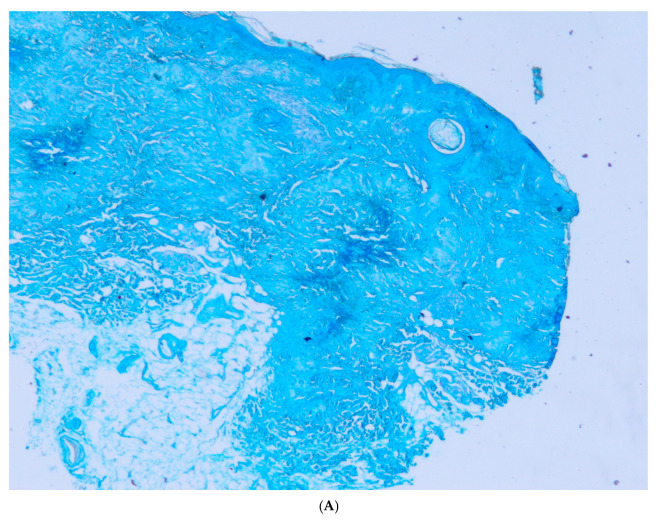
(**A**) Colloidal iron stain. Abundant mucin is evident in the center of granulomas. One of the granulomas is close to an incipient infundibular cyst (×20). (**B**) Verhoeff–van Gieson stain. The elastic fibers are evident in the papillary dermis but disappear in the granulomas except in their center, where they have a faint hue (×20). (**C**) Verhoeff–van Gieson stain. Detail of the transitional zone where the elastophagocytosis via multinucleated giant cells is evident (×200). (**D**) Verhoeff–van Gieson stain. Palisading granuloma in close relationship with a cyst, whose wall shows transepithelial elimination of elastic fibers (×20). (**E**) Verhoeff–van Gieson stain. Detail of the transepithelial elimination of elastic fibers (×100).

**Table 1 dermatopathology-12-00016-t001:** Cases of granuloma annulare with cysts and comedones. Some epidemiologic, clinical and histopathologic aspects.

Reference/ Year of Publication	Age/Sex	Location of Lesion	Elastophagocytosis/ Type of Multinucleated Giant Cell	Presence of Comedones/ Cysts	Transepithelial Elimination of Elastic Fibers	Associated Diseases	Treatment ***
Sudy et al., 2006 [26]	57/F	Forearms	YES/NA	YES/NO	NO	DM	DM therapy
Bhushan et al., 2011 [27]	42/F	Chest *, forearms, forehead	YES/NA	YES/NO	NO	Hypothyroidism DM	DM therapy
Gavioli et al., 2017 [28]	64/M	Forearms	YES/Langhans-type multinucleated cells **	YES/YES	NO	DM	NA
Pensler et al., 2024 [29]	57/M	Hands, forearms, elbows	NA/NA	YES/NO	NO	Insulin-dependent type II DM	NA
Our case	71/F	Elbows	YES/Langhans-type multinucleated cells	YES/YES	YES	Uncontrolled DM, AH, Hypercholesterolemia	Intralesional steroids

AH: arterial hypertension; DM: diabetes mellitus; F: female; M: male; NA: not available. * The chest lesions had no comedones. Instead, the sun-exposed lesions contained comedones. ** The type of multinucleated giant cell has been disclosed using histopathologic pictures because it is not specifically explained in the text. *** Indicates the treatment that improved GA.

## Data Availability

No further data are available.

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
