# Peer review of "Perforating Granuloma Annulare with Cysts and Comedones"

_dermatopathology, 2025, doi:10.3390/dermatopathology12020016_

Round 1
Reviewer 1 Report
Comments and Suggestions for Authors
- Title: Cystic granuloma annulare or actinic granuloma annulare in association with comedones and cysts? This is not the same, both semantically and histopathologically.
- Missing special stains: Granuloma multiforme Leiker (1964) was a widely observed and honored diagnosis from the early 1960s (foremost in the global South) that we must take as an important reminder of infectious diseases (in particular leprosy and deep fungal infections) camouflaging under the clinicopathological veil of granuloma annulare or any arciform palpable skin lesion. Most remarkably, in the present report the authors did not (sic!) stain for infectious organisms (PAS, Ziehl-Neelsen, Silver-Meth, specific immuno-stains) – neither did they stain for/investigate for possible foreign bodies (polarizing light etc.). These grave omissions cause serious misgivings! Remarkably, Fig.2B shows an intraepithelial micro-abscess, as often occurs in deep fungal and bacterial infections!
- Clinical picture and diagnosis: Fig.1B [relapse] looks quite dramatic to me; there is even a small satellite lesion adjacent the the large arciform red zone. I have serious qualms calling this clinical scenario “granuloma annulare”. Foremost, I would like to see the right-arm lesion, while the authors present only the left arm settings. Without the missing underpinnings of negative infectious-organisms-stains, this lesion is highly suspicious to haven an infectious origin.
- Diabetes mellitus: Apparently patients with that particular disorder share underlying diabetes mellitus as a common denominator. Treatment (at least in 2 cases) was reported in the literature. The authors seem to pay very little attention to the underlying diabetes mellitus in their patient – or do not report about it. Was intralesional steroid the only therapeutic endeavor?
- Interpretation of histopathological features: Elastophagocytosis may occur in virtually any granulomatous setting of that kind, and may be considered as unspecific. To paraphrase the late A.B.Ackerman: These histiocytes are in a feeding frenzy! The same may be true for the phenomenon of transepidermal elimination of cells and elastic/collagenous fibres (Fig.3D) and perforating inflammation as such. I would not take these phenomena as proof for ACTINIC granuloma annulare.
- Etiopathogenesis: In particularly one paragraph [lines 82-92] is far too sketchy and hard to comprehend. I suspect a variant of mumbo-jumbo – at least in line 85 where chromosome data (23p4) float around. In short, clearer writing would be highly appreciated.
Comments on the Quality of English Language
Typos, semantics, et al.: Not perfect yet. A fine brush would help!
Author Response
We appreciate very much your suggestions. The report is much better with the changes that you propose. We try to answer and to modify the report according to your recomendations.
We have done:
1.- We have changed the title to: “perforating granuloma annulare withs cysts and comedones”
2 and 3.
- we specifically emphasize that PAS, Gram and ziehl-neelsen stains were negative, as well as tha polirized light exploration. No pictures of these stains are provided but we have avaible if necessary.
- We add the clinical picture of the opposite elbow.
- We largely discuss why is granuloma annulare and not a fungal or mycobacterial infection.
4.- We add information about the blood levels of glucosa in the relapses, and in the non-relapsing time. However, we don’t have information enough about the evolution of the skin lesions matched with the evolution of her diabetes because the diabetes mellitus was controlled in another institution.
5.- We aggre with the reviewer. For us actinic granuloma annulare is a granuloma annulare. Some authors consider actinic granuloma annulare on basis of solar elastosis, presence of multinucleated giant cells and sometimes elastophagocytosis.
We may used both terms as synonims in the text and this could be confusing. We try to solved this misunderstanding.
In the introduction we stay:
“Actinic GA or annular elastolytic giant cell granuloma is a controversial condition, considered by us as a variant of GA. Actinic granuloma is characterized by lesions in sun-exposed areas2. Histopathologically, it shows the presence of multinucleated giant cells and is sometimes associated with elastophagocytosis, in addition to the typical changes of GA2. “
On the other hand, perforating is a sign of scratching, or elimination of foreign bodies or altered tissue. Perforating GA is considered an uncommon variant of GA, althoug probably the material transepithelial elimination is due to scratching and/or elimination of altered tisuue. In our case the transepithelial elimination of material occur in the bottom of a cyst, so it makes difficult explain via scratching. And probably is secondary to elimination of altered material. For us, It is a exceptional finding in a exceptional case. We add paragrah with this explanation.
6.- We completely rewrite the paragraph of etiopathogeny. We hope this time is understable.
7.- The english has been reviewed by MPDI services

Reviewer 2 Report
Comments and Suggestions for Authors
A unifyng hypothesis seems more appropriate for these two conditions(GA) so the title should be Actinic cystic annular granuloma.The histological diagnosis is complete perhaps more clinical differential diagnosis are needed at first glance.Important to underline loss of follicular support and resultant of open comedones formation as in senile comedones.Why transepitelial elimination?
Author Response
We appreciate very much your suggestions. The report is much better with the changes that you propose. We try to answer and to modify the report according to your recomendations.
We have done:
We have changed the title to a more appropiate: “perforating granuloma annulare withs cysts and comedones”
In the differential diagnosis we have included some new entities, and make a deeper differntial with fungal or mycobacterial infections.
Besides, we try to explain the transepithelial elimination of material.
Reviewer 3 Report
Comments and Suggestions for Authors
This article describes a case of granuloma annulare (GA) coexisting with adjacent cysts and comedones. This association seems rare and worth reporting. However, there are several drawbacks with this paper :
General comments :
- The title sounds inadequate : GA is a dermal, connective-tissue lesion, and as such it cannot be ‘cystic’ (cysts refer to cavities lined by an epithelium). A more appopriate term would be ‘Granuloma annulare with cysts and comedones’, or ‘Perforating Granuloma annulare with cysts and comedones’.
- The English needs thorough revision, namely for typos (eg ‘Langahn’s’, ‘yellowish-orangish structureless associated with white …’, ‘granulmas’, ‘dissapear’…, ‘this association25-28 4-7.’- line 127, ‘second biopsy (Fig 2B).3 (line 89), ‘is evidente’, ‘Not avible’, ‘sindrome’ ….) and grammar/syntax errors (‘The granulomas were formed by … and centered by … which contains mucin). The sentence « A) The elastic fibers are evidente in papilar dermis, but dissapear in the granulmas but their center where they have a faint hue », besides being unintelligible, is written in mixed Spanish-English and peppered with typos. The sentence ‘Usually involving the dorsum of hands and feet, and elbows. » has no main verb. …’to Actinic_granuloma, so called O’brien… ‘ should read ‘..to actinic_granuloma, so called O’Brien…’, ‘Actinic GA’ should read ‘actinic GA’ (lines 128 & 145), ‘..the diagnosis of GA over other palisading granulomatosis’ should read ‘the diagnosis of GA over other palisading granulomas’ etc. Awkward/inadequate/erroneous sentences/expressions : ‘widely extended for the body’, ‘Individual consideration deserves to Actinic granuloma’ ‘Histopathologically. Our case..’. The quotation marks in ‘M1’ and ‘M2’ are unnecessary
Specific comments:
- The description of the case could be more detailed. Did the patient have a past history of heavy sun-exposure ? what was her skin type ? what treatment was she taking for the associated diseases ? was she treated for diabetes ? if so, whzat was the effect of the treatment on the lesions (reportedly other similar cases healed after appropriate antidiabetic treatment).
- A photomicrograph of alcian blue staining should be provided to show mucin deposits, which is an important diagnostic clue of GA. Futhermore, I suggest to presnt results of PAS and Ziehl-Neelsen (or Fite) stain to rule out an (associated) fungal or mycobacterial infection.
- Ostia are not really seen in fig. 1A – as far as I can see, only milia-like microcysts are seen
- Panels 2B and 2C seem to have been inverted, considering the description provided in the corresponding legends
- 3C is of low quality (many artefacts) and should be replaced by a better-quality one
- References 31-33 concern the same book (Kerl H, et al. Diagnostic cutaneous pathology. Clinical-pathological correlation of inflammatory and other non-neoplastic skin diseases. A textbook and atlas), albeit (apparently, according to the pages) different chapters. Each one of them should include the title of the chapters it refers to. Besides, the references are not presented in a uniform style – please check and amend them appropriately. Line 89 : ‘Wang et al’ please cite the reference (9)
- The English needs thorough revision, namely for typos (eg ‘Langahn’s’, ‘yellowish-orangish structureless associated with white …’, ‘granulmas’, ‘dissapear’…, ‘this association25-28 4-7.’- line 127, ‘second biopsy (Fig 2B).3 (line 89), ‘is evidente’, ‘Not avible’, ‘sindrome’ ….) and grammar/syntax errors (‘The granulomas were formed by … and centered by … which contains mucin). The sentence « A) The elastic fibers are evidente in papilar dermis, but dissapear in the granulmas but their center where they have a faint hue », besides being unintelligible, is written in mixed Spanish-English and peppered with typos. The sentence ‘Usually involving the dorsum of hands and feet, and elbows. » has no main verb. …’to Actinic_granuloma, so called O’brien… ‘ should read ‘..to actinic_granuloma, so called O’Brien…’, ‘Actinic GA’ should read ‘actinic GA’ (lines 128 & 145), ‘..the diagnosis of GA over other palisading granulomatosis’ should read ‘the diagnosis of GA over other palisading granulomas’ etc. Awkward/inadequate/erroneous sentences/expressions : ‘widely extended for the body’, ‘Individual consideration deserves to Actinic granuloma’ ‘Histopathologically. Our case..’. The quotation marks in ‘M1’ and ‘M2’ are unnecessary
Author Response
We appreciate very much your suggestions. The report is much better with the changes that you propose. We try to answer and to modify the report according to your recomendations.
We have done:
1.- We have changed the title to a more appropiate: “perforating granuloma annulare withs cysts and comedones”
We add the Fitzpartick skin type of the patient. The patient had a marked solar elastosis, that can see in figures 2c and 3b. In addition, in figure 1b is easy to apreciate sundamaged skin, however, We don’t have information about her outdoor activities.
We add information about the blood levels of glucosa in the relapses, and in the non-relapsing time and the diabetes treatment. However, we don’t have information enough about the evolution of the skin lesions matched with the evolution of her diabetes because de diabetes mellitus was controlled in another institution.
2.- We add figure 3 A. It shows mucin deposits in the center of granulomas by coloidal iron.
3.- We add figure 1B. It shows a close-up of the lesion where ostia of comedones are more evidente.
4.- We have modified the legend of figures.
5.- We know about the low quality of figure 3D (figure 3C in the previous version of the report). We have repeated the cuts and stains, but we can’t get a better picture. We have keeped the figure in the report but if you considere is not good enough, we can eliminate. May figures 2C and 3E supply it.
6.- We checked the references.
7.- The english has been reviewed by MPDI services
Round 2
Reviewer 1 Report
Comments and Suggestions for Authors
none
Author Response
We appreciate your time in reviewing our manuscript.
Reviewer 3 Report
Comments and Suggestions for Authors
The English still needs some improvement (eg line 74: '....she had developed...' would better read '...she developed'; 'High blood pressure' would better read 'Arterial hypertension'
Comments on the Quality of English LanguageThe English still needs some improvement (eg line 74: '....she had developed...' would better read '...she developed'; 'High blood pressure' would better read 'Arterial hypertension'. Lines 202 and 279, no article ('the') is needed before 'transepithelial...'. Line 245 the sentence 'in sporotrichosis, the antecedent is usually traumatism....' would better read ''In sporotrichosis there is usually a history of traumatism....'. Line 264, 'granulomatosis' should read 'granulomas'; In table 1, 'No' should read 'NO' for consistency (column 5, line8).
Author Response
We appreciate your time in reviewing our manuscript.
We have improved the english according to your suggestions. We have eliminated the article “the” before transepithelial; altough, we have kept in some sentences because we consider it necessary in selected sentences